# Assessing Dietary Exposure Risk to Food Preservatives Among the Eating-Out Population in Taiwan Using the Total Diet Study Method

**DOI:** 10.3390/foods14030365

**Published:** 2025-01-23

**Authors:** Hao-Hsiang Ku, Shih-Cheng Yang, Huai-An Hsiao, Jui-Sheng Chen, Min-Pei Ling

**Affiliations:** 1Institute of Food Safety and Risk Management, National Taiwan Ocean University, 2 Beining Road, Keelung City 202301, Taiwan; kuhh@mail.ntou.edu.tw; 2Department of Food Science, National Taiwan Ocean University, 2 Beining Road, Keelung City 202301, Taiwan; y770130@yahoo.com.tw (S.-C.Y.); schwarm331@gmail.com (H.-A.H.); 3Graduate Institute of Applied Geology, National Central University, 300 Zhongda Road, Taoyuan City 320317, Taiwan; jschen@geo.ncu.edu.tw

**Keywords:** food preservatives, benzoic acid, sorbic acid, dehydroacetic acid, total diet study, eating out

## Abstract

In recent years, due to the rapid pace of urbanization and increasingly hectic modern lifestyles that leave little time for home cooking, more and more people prefer to dine at food stands, restaurants, or supermarkets due to convenience. This type of people are often called the eating-out population. The general public may have a concept that most of the food items consumed by people eating out are first prepared for storage by vendors and are likely to contain more food preservatives. Excessive exposure to benzoic acid (BA), sorbic acid (SA), and dehydroacetic acid (DHA), which account for the highest number of violations of the amount of preservatives permitted in food, may cause potential human health risk. The purpose of this study was to investigate the human health risks of consuming preservatives used in food among for Taiwanese people who eat out. We applied the total diet study (TDS) method to analyze the concentrations of BA, SA, and DHA in the food items frequently consumed when people dine outside. The hazard index in percent acceptable daily intake (%ADI) of BA and SA for four exposure groups classified by age were calculated. In high-intake consumers, the highest hazard index of BA was 2.5%ADI for the 6–9 years old age group of the eating-out population, which still fell within the acceptable risk range. In addition, the risk appeared to be decreasing year-on-year, which may be related to year-on-year improvements of the way food products are processed in the food industry.

## 1. Introduction

As Taiwan is a subtropical country with warm and humid weather, food preservatives are widely used to inhibit the growth of bacteria, yeasts, and molds, extend the shelf life of food, and reduce the possibility of food poisoning. Common antimicrobial preservatives include benzoic acid and its salts (BA), sorbic acid and its salts (SA), dehydroacetic acid (DHA), calcium propionate, and sodium nitrite. Their incorrect use in terms of method and dose can lead to the excessive addition of preservatives in foods, creating a hazard for the health of consumers [1]. The Taiwan Food and Drug Administration (TFDA) has strict and detailed regulations for the scope and amount of preservatives allowed in foods. However, according to post-market monitoring reports from regulatory agencies across Taiwan, surveys of commercially available foods in recent years have shown that BA and SA are more likely than other preservatives to be illegally added to food. BA occurs naturally in various fruits, including peaches, plums, strawberries, apples, and cinnamon. This compound serves as an effective preservative in food and beverage products, primarily functioning as an antimicrobial and antifungal agent [2]. The European Union has established regulatory guidelines permitting the use of benzoic acid across designated food categories, subject to strict adherence to maximum usage limits. Under Directive 95/2/EC, member states are required to implement comprehensive monitoring systems for food additive consumption and utilization, enabling the assessment of consumption patterns and regional variations across member states [3]. SA is a short-chain fatty acid with two unsaturated bonds. It is widely used in the food, beverage, cosmetic, and other industries. DHA is widely used as a preservative in food products due to its antimicrobial properties. It inhibits the growth of a range of bacteria (such as *Bacillus subtilis* and *Escherichia coli*.) molds (including *Aspergillus* and *Penicillium* species), and yeasts, thereby extending the shelf life of perishable products. It is particularly effective in low-moisture food items, sauces, and baked goods [4].

The Nutrition and Health Survey in Taiwan (NAHSIT) during 2013–2016 asked the respondents about “eating out” and the “number of meals” consumed as a basis for surveying people’s eating-out habits. The results showed that people in Taiwan eat out between one and two meals a day [5]. In Taiwan and many other countries, this eating-out population usually buys their food at food stalls, restaurants, or supermarkets for convenience. However, food from these places is often thought to contain more food preservatives to facilitate preservation and maintain its quality. Internationally, two types of violations with regard to the use of food preservatives are frequently encountered: the first type involves the illegal use of preservatives that should not be added to foods; the second type concerns the excessive use of preservatives. According to post-market monitoring reports, illegal use of BA and SA is often found in pickled products, meat products, and noodle products. In view of these considerations, individuals who eat out regularly may be exposed to excessive preservatives. Therefore, we selected the eating-out population to examine the health risks of eating out. Currently, there is limited international research on the eating-out population, with discussions only covering cholesterol intake [6], body mass index [7], body weight gain [8], food poisoning [9], and food hypersensitivity [10]. Therefore, through the results of this study, we can understand the potential risks of food preservative intake among the eating-out population and the main contributing food sources.

Many countries have used the total diet study (TDS) method to assess whether their populations are at risk from target contaminant hazards. The U.S. Food and Drug Administration (USFDA) first applied TDS in 1961 in a program monitoring radioactive contamination in food, using it to measure the dietary intake of specific analytes in national or regional populations [11] and evaluate their associated health risks. TDS, also known as a “market basket study”, has been used as a national monitoring research tool for food contamination and dietary exposure [12,13,14]. It is based on a national representative of food consumption for different subgroups of the general population. TDS can be used to determine the levels of various contaminants and nutrients present in foods and to estimate the public health risk due to chronic exposure to the presence of chemical substances. TDS representative foods are designed to measure the average intake of chemicals found in cooked or processed foods [15].

In recent years, various countries have conducted TDS to investigate food additives, assess dietary exposure, and evaluate health risks associated with their consumption. In the United States, the U.S. Food and Drug Administration [16] monitored artificial sweeteners, preservatives such as sodium benzoate and sorbic acid, and synthetic food dyes in processed foods as part of its TDS efforts. In Canada, Health Canada focused on preservatives (e.g., nitrites and sulphites), colorants, and artificial sweeteners in beverages and snacks [17]. In France, ANSES assessed exposure to food additives including annatto (E160b), nitrites (E249–250), sulphites (E220–228), and tartaric acid (E334), particularly in processed and ready-to-eat foods [18,19]. Similarly, Austria analyzed dietary exposure to preservatives like sulphites, benzoic acid, and sorbic acid to estimate health risks [1]. In South Korea, investigations emphasized the dietary intake of benzoic acid and sorbic acid in commonly consumed foods [20]. Taiwan actively utilized the TDS sampling method to evaluate food additives, including benzoic acid, sorbic acid, and sulphites, assessing their presence in the population’s diet and potential health impacts [21,22,23]. Together, these studies aim to ensure compliance with regulatory standards like acceptable daily intake (ADI) and address public health concerns related to food additive consumption. Compared to previous food preservative risk assessment studies conducted in Taiwan [21,22,23], while the representative food items sampled and analyzed differ due to different study subjects, the experimental methods remain consistent, following the food preservative testing methods announced by the Ministry of Health and Welfare in Taiwan. In 2016, Taiwan conducted a TDS survey on food preservatives to assess the health risks associated with BA and SA dietary intake among general consumers in Taiwan. Based on eight exposure groups classified by age, the hazard index was evaluated. Among high-intake consumers, females aged 66 and above showed the highest risk from BA intake, with the 95th percentile at 61.7%ADI; for sorbic acid, males aged 3–6 years showed the highest risk, with the 95th percentile at 14.0%ADI [22].

The objectives of this study were 2-fold: (1) to analyze BA, SA, and DHA concentrations in representative foods according to the eating-out population’s dietary habits using the TDS method and (2) to assess the potential health risk of consuming food preservatives in ready-to-eat food among the eating-out population in Taiwan. The maximum concentration of each representative food was then matched with the consumption to obtain the estimated daily intake. Exposure estimates were calculated separately based on different consumptions of whole group (WG) or consumer only (CO) population. Moreover, we identify foods with greater contribution based on the risk assessment results, which can provide the main basis for priority management.

## 2. Materials and Methods

### 2.1. Representative Food List

In TDS studies of food additives conducted by various countries [1,16,17,18,19,20,21,22,23], representative samples primarily consist of commercially available processed foods. The sampling design for the core food list of this study was mainly based on the regulations for the specification, scope, application, and limitation of food additives in Taiwan. In addition, we added food items that are not in the regulations but may be added illegally by the industry. Based on the core food list, a representative food item list for the eating-out population was selected according to the following principles: (1) according to the NAHSIT database, respondents who ate out more than four meals a week (breakfast, lunch, and dinner) totaled 1151 people, and they were defined as the eating-out population, and (2) the food items without food intake data based on the NAHSIT database were screened out. We considered the top 50 food items with the highest consumption and highest frequency among respondents in each exposure group, where the sum of consumption for these top 50 foods accounted for more than 50% of the total food consumption in each exposure group.

### 2.2. Sample Preparation

This study follows the nationally used sampling method for risk assessment, called the total diet study method, also known as the “food basket” method [14]. The sampling approach involves dividing the country into multiple regions and conducting sampling in each area. The representative foods in this study were selected based on the eating-out population from the NAHSIT database, with a total of 50 samples collected. We divided Taiwan into four regions (North, Central, South, and East), and selected the city or county with the highest population in each region (New Taipei City, Taichung City, Kaohsiung City, and Hualien County). We sampled cooked foods in street vendors, local markets, or restaurants. After purchasing the various food items, we completed the sampling record in detail before visiting the nearest convenience store and using frozen delivery to return the samples to the laboratory for receiving, coding, weighing, pre-processing, preparation, and preservation. After completing the sampling, we categorized and mixed the samples. After removing the inedible parts, we homogenized, packed, and preserved the edible parts for analysis. All sampling and analyses were conducted from 2016 to 2017.

### 2.3. HPLC Analysis of Food Preservatives

The analysis of food preservatives was conducted using an HPLC system (Agilent 1260 Infinity II, Santa Clara, CA, USA) equipped with a photodiode array detector and a C18 chromatographic column (5 μm, 4.6 mm × 25 cm) (Avantor ACE C18-AR, Radnor, PA, USA). The procedure followed the food preservative testing protocol announced by the Ministry of Health and Welfare, Taiwan, with minor modifications. The analytical method was consistent with previous studies [22,23], ensuring reproducibility. The limit of quantification (LOQ) for food preservatives using this testing method is 0.02 g/kg.

Sample Preparation: A 15 g aliquot of homogenized test sample was mixed with 150 mL of 50% methanol solution. The mixture was subjected to ultrasonic vibration for 30 min. It was then diluted to 250 mL with 50% methanol solution. A portion of the extract was centrifuged at 3500 rpm for 10 min. The supernatant was filtered through a 0.45 μm membrane filter, and the filtrate was collected as the test solution.

HPLC Analysis: The HPLC analysis was carried out with a mobile phase flow rate of 1.0 mL/min. Detection wavelengths were set to 230 nm for BA and DHA, and 260 nm for SA. Details such as injection volume, column temperature, and autosampler temperature were not specified in the method. The mobile phase was prepared using a 10-fold diluted 5 mM citric acid buffer solution. This buffer was made by dissolving 7 g of citric acid and 6 g of sodium citrate in 1 L of deionized water and then filtering it through a 0.45 μm membrane. Methanol, acetonitrile, and the diluted buffer solution were mixed in a ratio of 1:2:7 (*v*/*v*/*v*), and the final mixture was filtered through a membrane filter to remove any impurities. Stock solutions of the preservatives were prepared by dissolving 125 mg of each reference standard in 5 mL of 0.1 N sodium hydroxide solution, which was made by dissolving 0.4 g of sodium hydroxide in deionized water to a final volume of 100 mL. The solutions were then diluted with deionized water to a final volume of 100 mL. Working standard solutions were prepared by diluting the stock solutions with 50% methanol solution to achieve concentrations ranging from 0.25 to 100 μg/mL.

The reagents used in this study included analytical-grade citric acid (J.T.Baker, Phillipsburg, NJ, USA) and sodium citrate (Thermo Fisher Scientific, Waltham, MA, USA) for buffer preparation, HPLC-grade methanol (Honeywell, Charlotte, NC, USA), acetonitrile (Honeywell, Charlotte, NC, USA), and deionized water with an electrical resistivity of over 18 MΩ·cm at 25 °C. The reference standards used for analysis were BA (AccuStandard, New Haven, CT, USA), SA (Sigma-Supelco, Darmstadt, Germany), and DHA (Dr. Ehrenstorfer, LGC Standards, Augsburg, Germany).

Identification Test and Content Determination: Preservatives in the samples were identified based on the retention times and absorption spectra of peaks obtained from the test solutions, which were compared with those of the corresponding reference standards. The quantification of preservative content was performed using a calibration curve generated from the working standard solutions. To ensure the accuracy and reliability of the analytical results, multiple quality control measures were implemented. Blank samples were analyzed to monitor potential contamination, while control samples were included to verify both the accuracy and precision of the method. Additionally, spiked samples were processed and analyzed to evaluate recovery rates, with the target recovery range set at 80–120%. Any samples exhibiting recovery rates outside this range were subjected to retesting to validate the findings.

### 2.4. Risk Characterization

The present study defined the respondents in the NAHSIT database who ate out more than four times a week (including breakfast, lunch, and dinner) as the eating-out population (total of 1151 respondents). We used observed individual means (OIM) to calculate each level of food consumption rate. Based on the food consumption rate and body weight parameters, we calculated the ratio of food intake with the same food item for each individual.

This study matched the maximum concentration of each representative food to consumption rate to obtain an estimated daily intake (EDI). The consumption rate is different according to the diet habits of different age groups. We divided the age groups of Taiwanese people into 6–9, 10–17, 18–65, and 65–80 years old. We then divided each age group into WG and CO populations. The average consumption rate of the WG population is the sum of consumption rate data among the respondents divided by the number of all respondents, whereas the average food intake of the CO population is the sum of the consumption rate data divided by the number of those respondents who had food intake (Table 1).

Combining the NAHSIT database, we used the TFDA’s regulations “Standards for Specification, Scope, Application and Limitation of Food Additives” to carry out food aggregation and coding. We created 17 major categories of 285 food items and used the SAS 9.4 software for data merging, calculating the number of people in each exposure group consuming the food, the body weight, and the consumption rate for each food item. Based on the exposure assessment, we listed the food items with an EDI of more than 5% of the total EDI as the main contributing food items.

For the calculations of detected concentrations, we referred to the Global Environment Monitoring System/Food Contamination Monitoring and Assessment Programme-European Region (GEMS/Food EURO) guidelines for food testing [24] to assess the concentration of food preservatives. If a large amount of the sample concentration data (more than 10–15%) was lower than the LOQ, all of the “not detected (ND)” or “not quantifiable (NQ)” results were assumed to be “0” and “limit of detection (LOD) or LOQ” to assess changes in the exposure assessment.

Based on the exposure assessment of each exposure group, we calculated the EDI (mg/kg-day) of preservatives in foods among each exposure group. The calculation is shown in Equation (1):(1)$${\mathrm{EDI}}_{ij}=\sum_{j=1}^{n}\frac{C_{i}\times {\mathrm{CR}}_{ij}}{{\mathrm{BW}}_{j}}$$
where C*_i_* is the concentration (mg/kg) of the analyte in food item *_i_*; CR*_ij_* is the consumption rate (g/person per day) of food item *_i_* by each exposure group *_j_*, and BW*_j_* is the average body weight of each exposure group *_j_*. In all EDI values, the upper-bound (UB) values were used as the worst-case scenario, even though there might be a slight overestimation in the total sums. UB concentrations were calculated on the assumption that all the values below the LOQ are replaced by the LOQ, while for lower-bound (LB) concentrations, all the values below the LOQ are replaced by zero.

We integrated all of the results from each step of the exposure assessment and the hazard characteristics to calculate the hazard index (% acceptable daily intake, %ADI) of BA and SA in the ready-to-eat food consumed by the exposure groups. When %ADI was greater than 100, this indicated that the intake of BA and SA may pose a health risk to the exposure population. We did not calculate the DHA hazard index in this study because there were no available ADI data for recommendations. The hazard index values expressed as %ADI were calculated according to the following equation (Equation (2)):(2)$$\sum_{i=1}^{n}\%{\mathrm{ADI}}_{ij}=\sum_{i=1}^{n}\frac{{\mathrm{EDI}}_{ij}}{{\mathrm{ADI}}_{p}}\times 100\%$$
where ADI*_p_* is the ADI of preservative p in the food item, and %ADI is the ratio of EDI*_ij_* and ADI*_p_*. The ADI values of BA and SA used in this study were 20 and 25 mg/kg-day, respectively [25,26]. Since there is no internationally recommended ADI value for DHA, this study does not calculate the health risk of DHA to human health.

## 3. Results and Discussion

### 3.1. Food Preservative Concentration Analysis

The total number of food samples was 94. A total of 50 mixed food samples were analyzed. The results of the food preservative analysis, following the completion of testing, are shown in Table 2. The ratio of BA was 8%. BA was only detected in pickled vegetables, tofu products, dried tofu products, and wet noodles, dough, and similar products, among 50 food samples. Pickled vegetables were found to have the highest content of 170 mg/kg, followed by tofu products. The ratio of SA was 14% with the highest level of 1100 mg/kg found in the food item of Western-style ham (cured, smoked, and boiled), followed by cake, (sweet) biscuits, and pies (fruit fillings or soufflés). The ratio of DHA was 2%. Two hundred mg/kg of SA was only detected in the food item of fat, cream, and blended spreads.

### 3.2. Estimated Daily Intake and Risk Assessment

BA exposure survey and risk assessment: The CO population (LB) EDI was in the range of 0.16–0.25 mg/kg-day. The CO population (UB) EDI was in the range of 0.4–0.68 mg/kg-day. The WG population (LB) EDI was in the range of 0.06–0.08 mg/kg-day. The WG population (UB) EDI was in the range of 0.29–0.51 mg/kg-day. The risk assessments were based on the exposure survey results, calculating %ADI values. The %ADI of the CO population (LB) was in the range of 3.3–5.1%. The %ADI of the CO population (UB) was in the range of 7.9–13.5%. The %ADI of the WG population (LB) was between 1.3 and 1.7%. The %ADI of the WG population (UB) was in the range of 5.7–10.2% (Figure 1A). Previous research conducted a TDS on the dietary habits of Taiwan’s population, showing that for males aged 19–50, the BA intake risk was 20%ADI at the 50th percentile and 34%ADI at the 95th percentile [22]. In a Korean study [19], the 90th percentile BA intake risk for high-consumption groups in Korea was 26.1%ADI, with beverages being the main contributors to BA exposure. Compared to previous studies, the significant reduction in BA risk in this study of the eating-out population is mainly due to the WHO/FAO-recommended ADI value being updated from 5 mg/kg bw-day to 20 mg/kg bw-day in 2021 [25]. Furthermore, comparing BA exposure levels from Ling et al.’s [26] previous study, adult males aged 19–50 had a BA exposure of 1.0 mg/kg bw-day at the 50th percentile and 1.7 mg/kg bw-day at the 95th percentile. In this study, BA exposure for males aged 19–65 in the eating-out population was from 0.08 mg/kg bw-day (WG population (LB)) to 0.52 mg/kg bw-day (CO population (UB)). The results of this study show that even when analyzing BA using a sampling list designed specifically for the eating-out population (a high-exposure group), the BA exposure levels still showed a significant decrease. It is speculated that due to increased public awareness of food safety and advances in the food industry, BA exposure levels have also notably decreased.

SA exposure survey and risk assessment: The CO population (LB) EDI was in the range of 0.51–1.77 mg/kg-day. The CO population (UB) EDI was in the range of 0.75–2.21 mg/kg-day. The WG population (LB) EDI was in the range of 0.09–0.52 mg/kg-day. The WG population (UB) EDI was in the range of 0.33–0.96 mg/kg-day. The risk assessments were based on the exposure survey results, calculating %ADI values. The %ADI of the CO population (LB) was between 2.1 and 7.1%. The %ADI of the CO population (UB) was between 3.0 and 8.8%. The %ADI of the WG population (LB) was between 0.4 and 2.1%. The %ADI of the WG population (UB) was between 1.3 and 3.8% (Figure 1B). Ling et al. [18] indicated that for males aged 19–50, the SA intake risk was 4.1%ADI at the 50th percentile and 8.0%ADI at the 95th percentile. All these risks were below 100%ADI and were therefore considered acceptable. For the high-consumption groups in Korea, the 90th percentile SA intake risk was 8.4%ADI, with soy sauce, processed fish products, and kimchi being the main contributors to sorbic acid exposure.

DHA exposure survey: The results of the sample analysis showed that DHA was only detected in fat, cream, and blended spreads. The concentration was 200 mg/kg. For the adolescent CO population, EDI was in the range of 0.02–0.01 mg/kg-day, which was the highest of all age groups. The elderly group did not ingest any food products with DHA.

### 3.3. Contribution of Major Food Groups

The main food items contributing to BA exposure (≥5%) differ across age groups. For children aged 6–9 years, the primary contributors are wet noodles, dough, and similar products, accounting for 42%, followed by tofu products at 41%, pickled vegetables at 10%, and dried tofu products at 7%. For adolescents, tofu products contribute the most at 43%, followed by wet noodles, dough, and similar products at 37%, pickled vegetables at 14%, and dried tofu products at 6%. For adults, wet noodles, dough, and similar products are the largest contributors at 40%, while tofu products account for 34%, pickled vegetables for 18%, and dried tofu products for 8%. For the elderly aged 65–80 years, tofu products contribute the most at 53%, followed by wet noodles, dough, and similar products at 26%, and pickled vegetables at 19%. Further details are illustrated in Figure 2A.

Furthermore, the main food items contributing to SA exposure (≥5%) vary across age groups. For children aged 6–9 years, the primary contributors are cake, (sweet) biscuits, and pies with fruit fillings or soufflés, accounting for 46%, followed by Western-style ham (cured, smoked, and boiled) at 22%, non-emulsified sauces at 18%, and surimi-based products at 13%. For adolescents, Western-style ham (cured, smoked, and boiled) contributes the most at 42%, followed by cake, (sweet) biscuits, and pies with fruit fillings or soufflés at 23%, surimi-based products at 23%, and non-emulsified sauces at 9%. For adults, cake, (sweet) biscuits, and pies with fruit fillings or soufflés are the largest contributors, accounting for 42%, followed by Western-style ham (cured, smoked, and boiled) at 28%, surimi-based products at 17%, and non-emulsified sauces at 10%. For the elderly aged 65–80 years, Western-style ham (cured, smoked, and boiled) is the primary contributor, accounting for 52%, followed closely by surimi-based products at 44%. Further details are illustrated in Figure 2B. Among the 50 samples, DHA was only detected in cream and blended spreads at 200 mg/kg (Table 2), which was the main contributing source.

In a previous study, the main food sources of benzoic acid intake for Taiwanese children aged 3–6 were found to be sausages and seasonings, while the main sources of sorbic acid intake were pork products and cakes [22]. That previous study was conducted from 2010 to 2011, and when compared with this study, it shows that as time has changed, manufacturers may have reduced their use of food preservatives in response to increased public awareness of food safety. The main contributing food sources have also changed.

### 3.4. Uncertainty Analysis

Based on the sources of uncertainty in exposure assessment proposed by USEPA [27], this study discusses the research limitations encountered in this study accordingly. After quantifying the risks, we also considered the sources of uncertainty in this study. These can be categorized into three major categories.

Scenario uncertainty: In the present study, food aggregation analysis for the eating-out population was carried out using the NAHSIT database. People who ate out more than four times a week (including breakfast, lunch, and dinner) were categorized as the eating-out population. The results showed that the eating-out population accounted for about 15% of the population. As this percentage excluded people who ate out fewer than four times a week, the size of the eating-out population may have been underestimated.

Parameter uncertainty: (1) The NAHSIT database includes reviews of diets over a single 24 h day. It also contains overestimations of the variance of food intake between individuals; however, this has not affected the average estimate. (2) In the NAHSIT database, it is evident that some respondents were not aware of what foods they consumed. The data may include unknown foods, which cannot be clearly categorized when clustering foods. (3) In this study, food samples were collected from the four regions of Taiwan based on the representative foods consumed by the groups, and equal proportions were sampled and homogenized. The results were for single food items containing food preservatives, which diluted the content of the food preservatives.

Model uncertainty: Uncertainty arises due to errors in the mathematical model. In the present study, the OIM algorithm was used to calculate the consumption rate. The OIM model cannot correct variation errors in relation to individual food intake, producing a larger distribution for food intake. However, the average should be reliable.

## 4. Conclusions

From the perspective of the intake of food preservatives, the health risks posed to the eating-out population from consuming food preservatives were measured using food samples. The results showed that, in terms of the %ADI of BA exposure, the highest level of CO exposure was experienced by children at 5.1%. In terms of the %ADI of SA exposure, the highest level of CO exposure was experienced by children at 7.1%. The %ADI values of BA and SA exposure in Taiwan were both less than 100% and were within the range of acceptable risk. The methods of collecting representative foods from four regions in Taiwan and performing a homogeneous mixing of the food items caused the dilution of the content of the preservatives in the food items from a particular region, leading to an underestimation of the risk in that region. Therefore, this study suggests that future research can reduce the four regions discussed in this study to a single region or county/city, so that the results can more accurately reflect the local risk. The results of this study show that children aged 6–9 who eat out have the highest risk under both BA and SA intake, possibly due to uncertainties related to their consumption of a wider variety of food items and lower body weight. Moreover, children aged 6–9 and adults aged 19–65 who eat out share the same main contributing foods for BA and SA exposure, suggesting that children’s diets are influenced by their primary caregivers. Therefore, to reduce children’s BA and SA intake from eating out, primary caregivers need to pay more attention to the sources of children’s food. Regarding BA intake, since noodle shops are common in Taiwanese food stalls, and Taiwanese people’s dietary habits include consuming more wet noodles, dough, and similar products and tofu products, these two categories account for approximately 70% of total BA intake across all age groups. However, due to the low risk, there are no concerns about safety. Additionally, compared to previous research results [22], particularly given that this study’s eating-out population is a high-exposure group, findings show a declining trend in the risk of food preservative intake among the Taiwanese population. It is speculated that with advances in food processing technology, manufacturers have reduced the use of food preservatives in response to increased public awareness of food safety, so the public need not be concerned about excessive intake of food preservatives.

## Figures and Tables

**Figure 1 foods-14-00365-f001:**
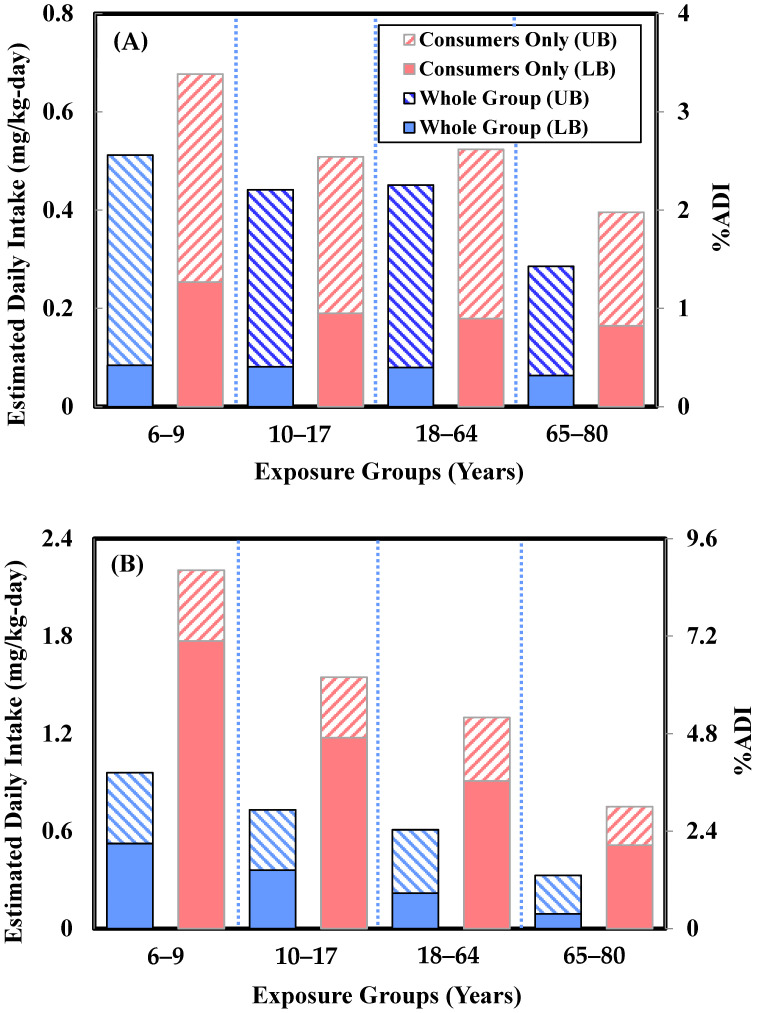
Comparison of the exposure risks of the eating-out population to (**A**) benzoic acid and (**B**) sorbic acid.

**Figure 2 foods-14-00365-f002:**
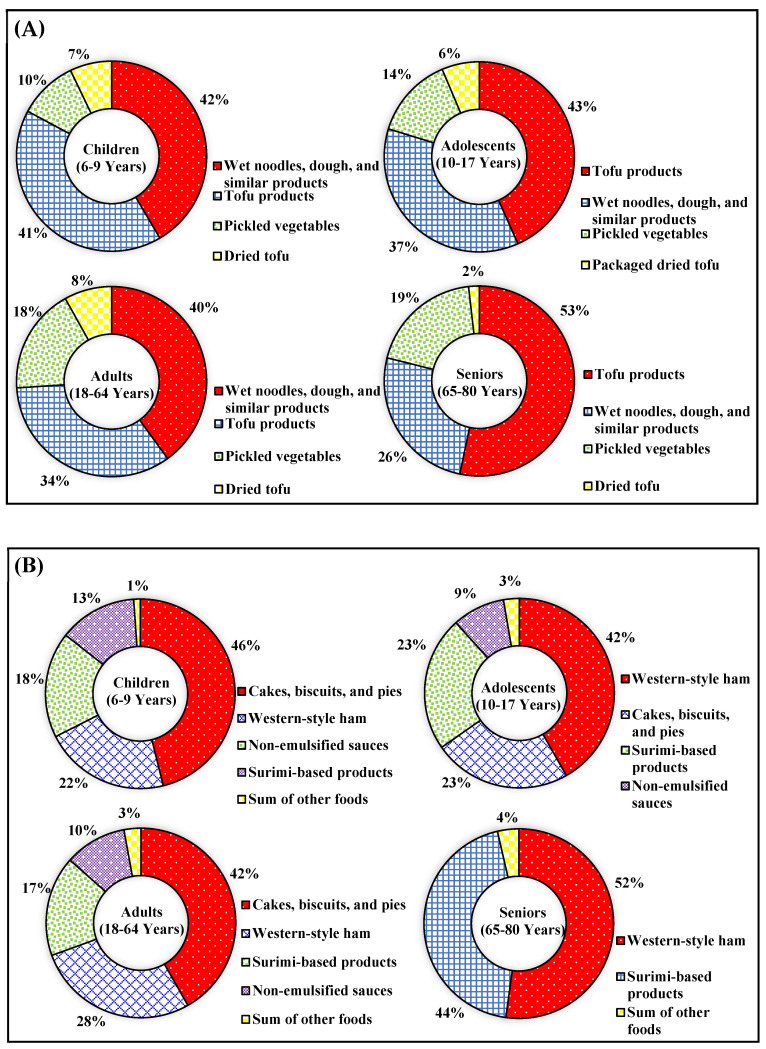
Contributions of food items to the exposure of the eating-out population to (**A**) benzoic acid and (**B**) sorbic acid.

**Table 1 foods-14-00365-t001:** Consumption rate (g/day) of representative foods by the eating-out population (g/day).

50 Food Items	Children (6–9 Years Old)		Adolescents (10–17 Years Old)		Adults (18–64 Years Old)		Seniors (65–80 Years Old)	
	Consumer Only (CO)	Whole Group (WG)	Consumer Only (CO)	Whole Group (WG)	Consumer Only (CO)	Whole Group (WG)	Consumer Only (CO)	Whole Group (WG)
Animal fat	1	0	2	0	1	0	1	0
Bean filling	64	1	57	3	85	9	29	1
Blended animal and plant fat	0	0	3	0	1	0	0	0
Breakfast cereal	78	3	154	7	159	13	62	16
Butter	1	0	2	0	6	0	0	0
Cakes, biscuits, and pies	90	15	105	10	103	12	0	0
Canned vegetables	27	8	31	9	28	4	39	4
Carbonated drinks	328	39	412	37	489	39	0	0
Concentrated soup products	41	6	11	10	8	7	2	2
Coffee beans, tea leaves, and grains for brewing	17	1	105	2	66	4	326	33
Cream and blended spreads	0	0	4	0	1	0	0	0
Creamer	12	0	49	2	25	1	12	1
Cured unheated minced meat	51	6	56	5	69	5	0	0
Deep-fried meat	67	4	115	14	110	9	39	2
Dried meat	31	3	35	3	24	1	41	6
Dried tofu	19	3	39	6	48	8	30	1
Dried or concentrated egg products	41	2	44	3	45	4	0	0
Dry noodles	81	14	101	14	181	29	268	27
Emulsified sauces	6	2	14	1	21	0	0	0
Fermented microbial products	2	0	27	17	17	10	6	3
Fermented soybean sauce	3	1	3	1	10	1	1	0
Frozen minced meat (Pork meatball)	59	10	65	9	58	6	27	1
Fried flour	31	1	54	2	60	2	88	9
Fruit puree drinks	255	17	336	23	358	34	49	2
Milk	309	45	331	67	408	43	216	22
Milk beverage	244	69	342	36	218	24	356	53
Milk powder	46	1	36	1	34	2	51	8
Non-carbonated drinks	444	205	686	499	903	675	644	290
Non-emulsified sauces	12	8	15	5	13	4	3	0
Other baked products	71	11	102	17	103	9	105	16
Other processed foods	218	174	369	264	386	261	300	180
Other soy sauce	10	5	15	13	17	14	12	10
Pickled vegetables	21	1	29	4	26	5	15	5
Pre-cooked noodles and dough	150	13	167	18	247	18	0	0
Protein products other than soybean	54	3	8	4	8	4	6	4
Sauce and braised meat products	37	3	61	5	33	7	11	2
Seasoning	3	2	41	3	45	4	25	3
Snacks made from potato, flour, or starch	55	13	53	14	76	15	8	1
Soy sauce	12	11	6	1	3	1	96	43
Soymilk	294	28	361	41	449	65	568	85
Starch-based snacks such as cereals, tuber root, and tuber	99	16	79	11	195	16	240	24
Steamed buns	93	10	151	18	187	25	79	8
Surimi-based products	48	7	71	16	60	8	88	9
Tofu	59	10	80	20	80	17	87	22
Tofu skin	18	1	44	3	63	3	0	0
Tuber root, tuber starch	11	5	11	5	11	2	0	0
Vegetable fat	12	12	12	12	17	15	15	14
Wet noodles, dough	104	34	129	55	151	67	139	35
Western-style ham	27	3	42	8	38	3	28	3
Yeast-fermented bread	70	18	100	40	101	24	81	12

**Table 2 foods-14-00365-t002:** Concentrations detected in each food category (mg/kg).

50 Food Items	*n*	Concentration (mg/kg)		
		BA	SA	DHA
Animal fat	1	ND	ND	ND
Bean filling	3	ND	ND	ND
Blended animal and plant fat	1	ND	ND	ND
Breakfast cereal	4	ND	ND	ND
Butter	1	ND	ND	ND
Cakes, biscuits, and pies	1	ND	480	ND
Canned vegetables	1	ND	ND	ND
Carbonated drinks	2	ND	ND	ND
Concentrated soup products	1	ND	ND	ND
Coffee beans, tea leaves, and grains for brewing	3	ND	ND	ND
Cream and blended spreads	2	ND	370	200
Creamer	1	ND	ND	ND
Cured unheated minced meat	1	ND	ND	ND
Deep-fried meat	3	ND	30	ND
Dried meat	1	ND	ND	ND
Dried tofu	1	50	ND	ND
Dried or concentrated egg products	1	ND	ND	ND
Dry noodles	2	ND	ND	ND
Emulsified sauces	1	ND	ND	ND
Fermented microbial products	1	ND	ND	ND
Fermented soybean sauce	1	ND	60	ND
Frozen minced meat (Pork meatball)	1	ND	ND	ND
Fried flour	1	ND	ND	ND
Fruit puree drinks	4	ND	ND	ND
Milk	3	ND	ND	ND
Milk beverage	4	ND	ND	ND
Milk powder	2	ND	ND	ND
Non-carbonated drinks	3	ND	ND	ND
Non-emulsified sauces	3	ND	350	ND
Other baked products	1	ND	ND	ND
Other processed foods	1	ND	ND	ND
Other soy sauce	1	ND	ND	ND
Pickled vegetables	2	170	ND	ND
Pre-cooked noodles and dough	1	ND	ND	ND
Protein products other than soybean	4	ND	ND	ND
Sauce and braised meat products	1	ND	ND	ND
Seasoning	1	ND	ND	ND
Snacks made from potato, flour, or starch	4	ND	ND	ND
Soy sauce	1	ND	ND	ND
Soymilk	1	ND	ND	ND
Starch-based snacks such as cereals, tuber root, and tuber	2	ND	ND	ND
Steamed buns	3	ND	ND	ND
Surimi-based products	4	ND	290	ND
Tofu	2	100	ND	ND
Tofu skin	1	ND	ND	ND
Tuber root, tuber starch	2	ND	ND	ND
Vegetable fat	3	ND	ND	ND
Wet noodles, dough	3	30	ND	ND
Western-style ham	1	ND	1100	ND
Yeast-fermented bread	1	ND	ND	ND

ND = not detected; detection limit = 0.02 g/kg.

## Data Availability

The original contributions presented in the study are included in the article, further inquiries can be directed to the corresponding author.

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
