# Peer review of "Assessing Dietary Exposure Risk to Food Preservatives Among the Eating-Out Population in Taiwan Using the Total Diet Study Method"

_foods, 2025, doi:10.3390/foods14030365_

Round 1

Reviewer 1 Report

Comments and Suggestions for Authors

Dear Authors,

The article you have published (doi: https://doi.org/10.1021/jf503987y) shares the same objective as the one outlined in this manuscript. Therefore, the main comment is to include novelty in your research.

Author Response

Responses to Reviewers Comments (Manuscript ID: foods-3335130)

We thank the reviewers’ valuable comments. These comments greatly improve the quality of this manuscript. The manuscript has been revised with the comments carefully considered. We used the blue markings to highlight the changes and new entries in the text, respectively. The following are the detailed responses to each individual comment.

Response to Reviewer #1's Comments:

  1. The article you have published (doi: https://doi.org/10.1021/jf503987y) shares the same objective as the one outlined in this manuscript. Therefore, the main comment is to include novelty in your research.

Response:

Thank you for your kind reminder. The novel purpose of this study is driven by the increasing global trend of eating out, and the common sense that street vendors, local markets, or restaurants often add more preservatives for convenient food storage. For this reason, this study has sampled and analyzed preservative residues in foods from street vendors, local markets, or restaurants, and verified whether these preservative residue levels pose any risks to human health. The previous research (Ling et al., 2015) aimed to assess health risks from food preservative consumption through various food types among the general population, while this study focuses on health risks from food preservative consumption through street vendor foods and restaurants among the eating-out population (a potentially high-exposure group), with completely different food sampling sources. We have revised the research objectives, representative foods list (Section 2.1), and sample preparation and analysis (Section 2.2) to avoid any misunderstanding.

Reviewer 2 Report

Comments and Suggestions for Authors

The issue raised concerns exposure to preservatives in food consumed by eating-out people in Taiwan. It is certainly very important and of interest to the potential reader. However, the paper in its current form has many shortcomings that should be improved.

Despite the announcement in the title that the study will refer to the total diet, no such mention is made in the paper. I therefore propose to shorten the title:

Assessing Dietary Exposure Risk to Food Preservatives Among Eating-Out Group in Taiwan

In Section 3 Results and Discussion, no studies other than their own were referred to, so there was not really a scientific discussion. It is important to find and point out at least some studies that indirectly deal with people eating-out.

In subsection 3.3, sentence equivalents are used. I believe they should be turned into sentences.

Lines 228-246 outline three types of ‘uncertainty’, as they are called. In my opinion, this word can be used for laboratory research. Here, however, the word ‘limitations’ should be used.

In the Conclusions, the words ‘female’ and ‘male’ are given, while the paper does not indicate gender-related results. Thus, the conclusions do not match the results, which is a significant flaw.

References in the text are not given as required by the Journal. They should be given in square brackets and numbered according to the order in which they appear in the text, and given in that order in the References. In addition, there are references in the References that are missing in the text and vice versa, which is a very serious deficiency .

For consideration:

Please consider whether including a value of 0 in Table 1 is necessary.

Please consider whether the food items in Table 1 and Table 2 could be listed in alphabetical order, which would make it easier to view these tables.

The drawings and descriptions in Figure 2 are not very clear. Perhaps they could be improved.

I also provide the following editorial comments:

13 group > people

19 food preservatives among > preservatives used in food among for

45 or > and

46-47 food stalls, restaurants or supermarkets > these places

146 (incorrect font for the letter ‘i’)

149 are > were

154 is > was

155 indicates > indicated

Table 2 (the font is larger than in Table 1 and should be reduced in size)

Figure 1 (the font size for the description of the X-axis on panels A and B is different and should be unified)

206 06.4.1 (???)

207 04.2.2.10.4 (???)

225 3.4. Uncertainty Analysis (???)

Author Response

Responses to Reviewers Comments (Manuscript ID: foods-3335130)

We thank the reviewers’ valuable comments. These comments greatly improve the quality of this manuscript. The manuscript has been revised with the comments carefully considered. We used the blue markings to highlight the changes and new entries in the text, respectively. The following are the detailed responses to each individual comment.

Response to Reviewer #2's Comments:

  1. The issue raised concerns exposure to preservatives in food consumed by eating-out people in Taiwan. It is certainly very important and of interest to the potential reader. However, the paper in its current form has many shortcomings that should be improved.

Response:

Thank you for the reviewer's valuable suggestions. We have made revisions according to your suggestions below.

  1. Despite the announcement in the title that the study will refer to the total diet, no such mention is made in the paper. I therefore propose to shorten the title:

Assessing Dietary Exposure Risk to Food Preservatives Among Eating-Out Group in Taiwan

Response:

Thank you for your suggestion. Since this study's sampling method uses a nationwide total diet study approach that divides the country into four regions - north, central, south, and east, we consider keeping TDS in the title, but we have revised and emphasized in the main text that sampling was conducted using the total diet study method. We have added and emphasized the total diet survey in the Section 1 (research objectives) and Section 2.1 (representative foods list) in the main text.

  1. In Section 3 Results and Discussion, no studies other than their own were referred to, so there was not really a scientific discussion. It is important to find and point out at least some studies that indirectly deal with people eating-out.

Response:

In current international literature, there is no discussion about the risks of food preservative intake among the eating-out population. Current discussions only cover cholesterol intake, Body Mass Index, body weight gain, food poisoning, and food allergy, which we have added explanations for in “Introduction” Section. Additionally, we have included comparisons and discussions between our previous research (Ling et al., 2015) and this study, which has been added to the “Results and Discussion” section and “Conclusion” section.

  1. In subsection 3.3, sentence equivalents are used. I believe they should be turned into sentences.

Response:

I have revised Section 3.3 into sentence format. Thank you for the suggestion.

  1. Lines 228-246 outline three types of ‘uncertainty’, as they are called. In my opinion, this word can be used for laboratory research. Here, however, the word ‘limitations’ should be used.

Response:

USEPA (2011) mentions three types of uncertainties in exposure assessment: (1) scenario uncertainty, (2) parameter uncertainty, and (3) model uncertainty, which is why we present it in Section 3.4 as "Uncertainty Analysis". We have added USEPA (2011) in the main text (Section 3.4) to explain and mention that this is also a research limitation.

  1. In the Conclusions, the words ‘female’ and ‘male’ are given, while the paper does not indicate gender-related results. Thus, the conclusions do not match the results, which is a significant flaw.

Response:

We apologize for the oversight. At the beginning of our research, we performed calculations by gender (please refer to Tables 1 and 2), but after finding no gender differences, we presented the results without gender differentiation. We have corrected this in the Conclusion section.

Table 1. Risk assessment results of benzoic acid exposure in the eating-out population

Age group

EDI
(mg/kg-bw/day)

%ADI

Consumer Only

Whole Group

Consumer Only

Whole Group

Male

0.23

0.08

4.5

1.6

6-9 years

Female

0.29

0.09

5.8

1.8

All

0.25

0.08

5.1

1.7

Male

0.19

0.08

3.9

1.6

10-17 years

Female

0.19

0.08

3.7

1.6

All

0.19

0.08

3.8

1.6

Male

0.18

0.08

3.7

1.7

18-64 years

Female

0.17

0.08

3.5

1.5

All

0.18

0.08

3.6

1.6

Male

0.16

0.06

3.2

1.1

65-80 years

Female

0.20

0.12

3.9

2.4

All

0.16

0.06

3.3

1.3

Table 2. Risk assessment results of sorbic acid exposure in the eating-out population

Age group

EDI
(mg/kg-bw/day)

%ADI

Consumer Only

Whole Group

Consumer Only

Whole Group

Male

2.26

0.48

9.0

1.9

6-9 years

Female

1.98

0.41

7.9

1.7

All

1.77

0.52

7.1

2.1

Male

1.04

0.37

4.2

1.5

10-17 years

Female

1.18

0.35

4.7

1.4

All

1.18

0.36

4.7

1.4

Male

0.75

0.15

3.0

0.6

18-64 years

Female

1.11

0.31

4.4

1.2

All

0.91

0.22

3.6

0.9

Male

0.52

0.10

2.1

0.4

65-80 years

Female

0.00

0.00

0.0

0.0

All

0.51

0.09

2.1

0.4

  1. References in the text are not given as required by the Journal. They should be given in square brackets and numbered according to the order in which they appear in the text, and given in that order in the References. In addition, there are references in the References that are missing in the text and vice versa, which is a very serious deficiency.

Response:

Thank you for your reminder. We have carefully checked and revised the references in the text.

  1. For consideration:

Please consider whether including a value of 0 in Table 1 is necessary.

Response:

We have removed the value of 0 from Table 1.

  1. Please consider whether the food items in Table 1 and Table 2 could be listed in alphabetical order, which would make it easier to view these tables.

Response:

Thank you for your suggestion. Originally, we categorized by food type, but due to space considerations in the layout, we removed the food type categories. We have revised Tables 1 and 2 to be in alphabetical order.

  1. The drawings and descriptions in Figure 2 are not very clear. Perhaps they could be improved.

Response:

We have updated Figure 2.

  1. I also provide the following editorial comments:

13 group > people

19 food preservatives among > preservatives used in food among for

45 or > and

46-47 food stalls, restaurants or supermarkets > these places

146 (incorrect font for the letter ‘i’)

149 are > were

154 is > was

155 indicates > indicated

Table 2 (the font is larger than in Table 1 and should be reduced in size)

Figure 1 (the font size for the description of the X-axis on panels A and B is different and should be unified)

206 06.4.1 (???)

207 04.2.2.10.4 (???)

225 3.4. Uncertainty Analysis (???)

Response:

Thank you for your suggestions. We have made all the revisions accordingly.

Reviewer 3 Report

Comments and Suggestions for Authors

Review report “foods-3335130”

In the study entitled "Assessing Dietary Exposure Risk to Food Preservatives Among Eating-Out Groups in Taiwan" the authors evaluated the health risks associated with exposure to benzoic acid (BA), sorbic acid (SA), and dehydroacetic acid (DHA) from food preservatives among individuals who frequently eat out in Taiwan. It employs a Total Diet Study (TDS) approach for sampling, analysis, and risk assessment.

While the study addresses an important topic, there are several limitations that should be addressed. The manuscript appears to be in a draft stage, and the study methodology and presentation can benefit from refinement.

Here the key issues:

Introduction: this section lacks a detailed discussion of the toxicological profiles and health implications of BA, SA, and DHA. Only few references, which are reported in wrong style.

In general, the background and the references are too few.

Exposure assessment study: The reliance on the Nutrition and Health Survey in Taiwan (NAHSIT), which is conducted in Chinese, limits the accessibility and replicability of the research to an international audience. Moreover, for the reviewers it is impossible to verify the “goodness” of data.

Limited number of samples analyzed: Only 94 samples were analyzed, which seems insufficient given the diverse dietary habits and regional food variations across Taiwan. This limited sample size may undermine the generalizability of the findings.

Samples analysis: describe with more details the methods for preservatives determination (are they validated? Introduce some references.)

Left censored data: there are too many left censored data lowering the “power” of the study.

Lack of “critical” discussion and comparison with other studies worldwide.

Minor issues

The manuscript contains some typos and grammatical errors, as well as structural inconsistencies that detract from its readability.

For these reasons, in my opinion, in the current form this study is not eligible for publication in Foods.

Comments on the Quality of English Language

Please see the review report

Author Response

Responses to Reviewers Comments (Manuscript ID: foods-3335130)

We thank the reviewers’ valuable comments. These comments greatly improve the quality of this manuscript. The manuscript has been revised with the comments carefully considered. We used the blue markings to highlight the changes and new entries in the text, respectively. The following are the detailed responses to each individual comment.

Response to Reviewer #3's Comments:

  1. Introduction: this section lacks a detailed discussion of the toxicological profiles and health implications of BA, SA, and DHA. Only few references, which are reported in wrong style. In general, the background and the references are too few.

Response:

Thank you for the reviewer's suggestions. We have added relevant introductions in Introduction Section. We also have added international food safety-related research on the eating-out population. Current researches only cover cholesterol intake, body mass index, body weight gain, food poisoning, and food hypersensitivity, which we have added explanations for in Introduction Section.

  1. Exposure assessment study: The reliance on the Nutrition and Health Survey in Taiwan (NAHSIT), which is conducted in Chinese, limits the accessibility and replicability of the research to an international audience. Moreover, for the reviewers it is impossible to verify the “goodness” of data.

Response:

Since 1993, Taiwan has been conducting the Nutrition and Health Survey in Taiwan, using 24-hour dietary recall questionnaires to collect annual food intake data across different age groups to assess nutritional changes in the population. This serves as a primary basis for various studies, such as: dietary patterns of Taiwanese older adults (Chuang et al., 2019) and the evaluation of type 2 diabetes incidence through food intake assessment (Tsai et al., 2019).

Chuang SY, Lo YL, Wu SY, Wang PN, Pan WH, 2019. Dietary patterns and foods associated with cognitive function in taiwanese older adults: the cross-sectional and longitudinal studies. Journal of the American Medical Directors Association. 20: 544.

Tsai TL, Kuo CC, Pan WH, Wu TN, Lin P, Wang SL, 2019. Type 2 diabetes occurrence and mercury exposure - From the National Nutrition and Health Survey in Taiwan. Environment International, 126: 260-267.

  1. Limited number of samples analyzed: Only 94 samples were analyzed, which seems insufficient given the diverse dietary habits and regional food variations across Taiwan. This limited sample size may undermine the generalizability of the findings.

Response:

We considered the top 50 food items (94 Samples) with the highest consumption and highest frequency among respondents in each exposure group, where the sum of consumption for these top 50 foods accounted for more than 50% of the total food consumption in each exposure group. We have added an explanation in Section 2.1. Please refer to Table 3.

Table 3. Representative food items for dining-out population

Item

Category/Type
(50 food items)

Food Samples
(94 Samples)

1   

Milk

Milk (brand 1)
Milk (brand 2)
Milk (brand 3)

2   

Milk beverage

Fermented Milk (Yakult)
Flavored Milk (Kuang Chuan)
Yogurt Drink (Uni-President)
Flavored Milk (Uni-President)

3   

Creamer

Creamer

4   

Milk powder

Milk Powder (brand 1)
Milk Powder (brand 2)

5   

Vegetable fat

Salad Oil (brand 1)
Salad Oil (brand 2)

6   

Animal fat

Lard

7   

Blended animal and plant fat

Shortening

8   

Butter

Butter

9   

Cream and blended spreads

Butter Spreads (brand 1)
Butter Spreads (brand 2)

10               

Pickled vegetables

Pickled Radish
Bamboo Shoots

11               

Canned vegetables

Corn Kernels

12               

Bean filling

Mung Beans
Red Beans
Chestnuts

13               

Soymilk

Soy Milk

14               

Tofu skin

Bean Curd Sheet

15               

Tofu

Traditional Tofu
Fried Tofu

16               

Dried tofu

Dried Tofu

17               

Tuber root, tuber starch

Sweet Potato Starch
Tapioca Starch

18               

Wet noodles, dough

Oil Noodles
White Noodles
Scallion Pancake Wrappers

19               

Dry noodles

Rice Noodles
Vermicelli

20               

Pre-cooked noodles and dough

Instant Fried Noodles

21               

Steamed buns

Meat Buns
Pan-Fried Buns
Plain Steamed Buns

22               

Fried flour

Fried Dough Sticks

23               

Breakfast cereal

Oatmeal (Quaker)
Oat Beverage (Quaker)
Five-Grain Powder
Wheat Powder (Quaker)

24               

Starch-based snacks such as cereals, tuber root, and tuber

Radish Cake
Tapioca Pearls

25               

Yeast fermented bread

White Bread

26               

Other baked products

Flavored Bread

27               

Cakes, biscuits, and pies

Cakes

28               

Cured unheated minced meat

Hot Dogs

29               

Frozen minced meat

Meatballs

30               

Sauce and braised meat products

Minced Pork Sauce

31               

Deep-fried meat

Fried Chicken Legs
Chicken Fillets
Fried Chicken Nuggets

32               

Western-style ham

Western-Style Ham

33               

Dried meat

Pork Floss

34               

Surimi-based products

Fish Cakes
Fish Paste
Oden
Fish Balls

35               

Dried or concentrated egg products

Braised Eggs

36               

Seasoning

Curry Sauce

37               

Concentrated soup products

Pork Broth

38               

Emulsified sauces

Mayonnaise
Salad Dressing

39               

Non-emulsified sauces

Ketchup
Mushroom Sauce
Black Pepper Sauce

40               

Fermented microbial products

Satay Sauce

41               

Fermented soybean sauce

Miso

42               

Soy sauce

Soy Sauce

43               

Other soy sauce

Thick Soy Sauce

44               

Protein products other than soybean

Gluten
Wheat Sausage
Wheat Rounds
Vegetarian Ham

45               

Fruit puree drinks

Orange Juice (Non-Fresh)
Lemon Juice (Non-Fresh)
Kiwi Juice
Mixed Fruit Juice

46               

Carbonated drinks

Cola (Coca-Cola)
Soft Drink (Sprite)

47               

Non-carbonated drinks

Black Tea
Milk Tea
Green Tea

48               

Coffee beans, tea leaves, and grains for brewing

Rice Bran
Sugar-Free Pouchong Tea
Milk Tea Pack (Lipton)

49               

Snacks made from potato, flour, or starch

Potato Chips (Lay’s)
Puffs (I-Mei)
Pringles
Rice Crackers (Want Want)

50               

Other processed foods

Clear Soup

  1. Samples analysis: describe with more details the methods for preservatives determination (are they validated? Introduce some references.)

Response:

Thank you for the reviewer's suggestions. We have added an explanation in the “Sample analysis” Section (2.2).

  1. Left censored data: there are too many left censored data lowering the “power” of the study.

Response:

The analytical method in this study is primarily based on the testing method announced by Taiwan's Food and Drug Administration. After extracting the samples through agitation, they were analyzed using HPLC. The Limit of Quantification (LOQ) for food preservatives in this testing method is 0.02 g/kg. Most of the analysis results were non-detectable, which this study interprets as indicating no food preservatives were added. However, when calculating potential human health risks in this study, the LOQ value (0.02 g/kg) was still used as the upper-bound (UB) concentration for conservative risk estimation.

  1. Lack of “critical” discussion and comparison with other studies worldwide.

Response:

We have included comparisons and discussions between our previous research (Ling et al., 2015) and this study, which has been added to the “Results and Discussion” section and “Conclusion” section.

Round 2

Reviewer 1 Report

Comments and Suggestions for Authors

Dear Authors,

Thank you for the revised manuscript and the detailed explanations regarding your previously published article (DOI: https://doi.org/10.1021/jf503987y). I appreciate your efforts to address the feedback provided. However, I have a few additional suggestions for further improvement:

·         Section 3.3: Contribution of Major Food Groups

There is no mention of DHA in this section. Please comment on the obtained results for detected DHA in cream and blended spreads. Additionally, which age group is primarily affected?

·         Lines 333–334:

The statement “Preservatives such as salicylic acid and p-hydroxybenzoate, which are banned in Taiwan, were not detected in the samples” should be removed from the Conclusion. New information should not be introduced in the Conclusion section. Instead, this data should be included in the main text.

·         Line 194:

Alongside BA and SA, DHA was also tested. Please ensure DHA is mentioned here.

·         Line 213:

Instead of SA, I assume the intended compound is DHA. Please confirm and correct if necessary.

·         Frozen Minced Meat (Table 1):

This is listed as one of the food items sampled. Please elaborate on how such products are typically obtained from street vendors, local markets, or restaurants and clarify whether they qualify as ready-to-eat foods (as mentioned on line 120: “We sampled cooked foods from street vendors, local markets, or restaurants”).

·         Lines 290–296:

This section provides a good starting point for discussing the results. Please expand with a more detailed comparison and critical analysis. Are the results consistent across the different age groups? Share your insights on whether the observed patterns align with expectations.

Author Response

Responses to Reviewers Comments (Manuscript ID: foods-3335130)

We thank the Reviewer's valuable comments, which have made this manuscript more complete. The manuscript has been revised with the comments carefully considered. We used the blue markings to highlight the changes and new entries in the text, respectively. The following are the detailed responses to each individual comment.

Response to Reviewer's Comments:

  1. Section 3.3: Contribution of Major Food Groups

There is no mention of DHA in this section. Please comment on the obtained results for detected DHA in cream and blended spreads. Additionally, which age group is primarily affected?

Response: Since DHA was only detected in cream and blended spreads at 200 mg/kg among the 50 samples, the major contributing foods section does not mention the main contributing foods for DHA. Based on the consumption of cream and blended spreads, the 10-17 age group has the highest intake, therefore the impact on consumer-only exposure is relatively greater for this age group. We provide further explanation in section 3.3.

  1. Lines 333–334:

The statement “Preservatives such as salicylic acid and p-hydroxybenzoate, which are banned in Taiwan, were not detected in the samples” should be removed from the Conclusion. New information should not be introduced in the Conclusion section. Instead, this data should be included in the main text.

Response: In addition to analyzing BA, SA, and DHA, this study also analyzed salicylic acid and p-hydroxybenzoate, which are banned in Taiwan. Since these were not detected, they were not discussed in the manuscript. To avoid misunderstanding, we have deleted this sentence. Thank you for the Reviewer's kind reminder.

  1. Line 194:

Alongside BA and SA, DHA was also tested. Please ensure DHA is mentioned here.

Response: Since there is no internationally recommended ADI value for DHA, it was not mentioned in the risk assessment calculations in this study. We have added this explanation to avoid misunderstanding.

  1. Line 213:

Instead of SA, I assume the intended compound is DHA. Please confirm and correct if necessary.

Response: We have confirmed this again and it is correct.

  1. Frozen Minced Meat (Table 1):

This is listed as one of the food items sampled. Please elaborate on how such products are typically obtained from street vendors, local markets, or restaurants and clarify whether they qualify as ready-to-eat foods (as mentioned on line 120: “We sampled cooked foods from street vendors, local markets, or restaurants”).

Response: According to the eating-out population's consumption in Taiwan, within the “frozen minced meat” category, “pork meatball” has the highest consumption, so we purchased this food item from restaurants across northern, central, southern, and eastern regions of Taiwan, then combined the samples for analysis. “Pork meatball” has been added in parentheses in Tables 1 and 2 to clarify the “frozen minced meat” category.

  1. Lines 290–296:

This section provides a good starting point for discussing the results. Please expand with a more detailed comparison and critical analysis. Are the results consistent across the different age groups? Share your insights on whether the observed patterns align with expectations.

Response: Thank you for the Reviewer's valuable suggestions. We have added more discussion in the conclusion chapter based on our research findings.

Reviewer 2 Report

Comments and Suggestions for Authors

The authors corrected the paper according to my comments or addressed them properly.

Author Response

We thank the Reviewer's valuable comments.

Reviewer 3 Report

Comments and Suggestions for Authors

Dear authors, 

unfortunately the work of revision done for the present manuscript is not sufficient. Many problems described in the previous review report remain. Among them:

- number of references should be substantially expanded as described by the reviewer in the comment box of references and reported in the correct style 

- methods are not adequately described. Moreover in all section M&M for all reagents, standards, software and so on there should be added (company, city, country)

- the preliminary part of discussion of the occurrence is not discussed 

- no comparison study with other relevant studies for analysis of these preservatives: (1) analytical approach used, (2) occurrence and monitoring studies (3) other TDS. The authors may focus their attention on last five years.

I have to repeat that the potency of the study is very limited due to some points (low samples number and high percentage of left censored data), so if also other things are not adequately developed the study and the manuscript remain only an exercise (draft stage).

I would like to suggest authors to take time and improve substantially the manuscript.

Author Response

Responses to Reviewers Comments (Manuscript ID: foods-3335130)

We thank the Reviewer's valuable comments, which have made this manuscript more complete. The manuscript has been revised with the comments carefully considered. We used the blue markings to highlight the changes and new entries in the text, respectively. The following are the detailed responses to each individual comment.

Response to Reviewer's Comments:

  1. number of references should be substantially expanded as described by the Reviewer in the comment box of references and reported in the correct style

Response: Thank you for your suggestions. We have added related research on international total diet studies and eating-out populations in the Introduction section, and discussed the results in comparison with previous literature.

  1. methods are not adequately described. Moreover in all section M&M for all reagents, standards, software and so on there should be added (company, city, country)

Response: Thank you for the Reviewer's comments. We have created a separate “2.3 Sample Analysis” section to explain the experimental analysis process.

  1. the preliminary part of discussion of the occurrence is not discussed

Response: We have added international studies on the eating-out population, with discussions only covering cholesterol intake (Sharmin et al., 2017), body mass index (Vincze et al., 2023), body weight gain (Bezerra et al., 2012), food poisoning (Worsfold, 2006), and food hypersensitivity (Knibb et al., 2024) in the Introduction section. Additionally, we have added the discussion in the “Introduction” section and “3.2 Estimated Daily Intake and Risk Assessment” section.

  1. no comparison study with other relevant studies for analysis of these preservatives: (1) analytical approach used, (2) occurrence and monitoring studies (3) other TDS. The authors may focus their attention on last five years.

Response: We have cited TDS surveys on food additives conducted in recent years by the United States (USFDA, 2020), Canada (Health Canada, 2020), France (ANSES, 2019; Bemrah et al., 2012), Austria (Mischek and Krapfenbauer-Cermak, 2012), and Korea (Shin et al., 2017) as examples. We have cited international TDS surveys on food additives conducted in recent years by the United States, Canada, France, Austria, and Korea as examples. We have also included comparisons with previous Taiwanese studies. All these TDS surveys purchased commercially processed foods and analyzed food additive concentrations using HPLC. We explain the related studies in the introduction section, and compare and explain the experimental analysis in sections 2.1 and 2.3.

  1. I have to repeat that the potency of the study is very limited due to some points (low samples number and high percentage of left censored data), so if also other things are not adequately developed the study and the manuscript remain only an exercise (draft stage).

Response: Thank you for the Reviewer's valuable comments. We have added detailed experimental procedures for this study in section 2.3. This experiment was conducted according to nationally announced testing methods.

  1. I would like to suggest authors to take time and improve substantially the manuscript.

Response: Thank you for the Reviewer's valuable comments. We have added detailed discussions of the literature and explanations of the experimental procedures.

Round 3

Reviewer 1 Report

Comments and Suggestions for Authors

Dear Authors,

Thank you for accepting the suggestions.

Author Response

We thank the Reviewer's valuable suggestions.

Reviewer 3 Report

Comments and Suggestions for Authors

After the second round of revisions, unfortunately I think even more that the manuscript is not suitable for publication in foods. Much information is summary, incorrect (e.g., reference to regulations and directives repealed years ago) and the reviews were conducted hastily, not adequately considering many points.

Author Response

We thank the Reviewer's valuable comments, which have made this manuscript more complete. We used the purple markings to highlight the changes and new entries in the text, respectively. We have added all reagents, standards, and equipment (including information on company, city, country) in the Materials and Methods section, and have rewritten the content of the "Identification Test and Content Determination". We also have rechecked and confirmed the accuracy of all cited regulations and directives. Thank you again for your constructive suggestions. We hope these revisions meet the requirements and improve the clarity and quality of the manuscript.